# Q Fever: Seroprevalence, Risk Factors in Slaughter Livestock and Genotypes of *Coxiella burnetii* in South Africa

**DOI:** 10.3390/pathogens10030258

**Published:** 2021-02-24

**Authors:** Maruping Mangena, Nomakorinte Gcebe, Rian Pierneef, Peter N. Thompson, Abiodun A. Adesiyun

**Affiliations:** 1Department of Production Animal Studies, Faculty of Veterinary Science, University of Pretoria, Private Bag X04, Onderstepoort, Pretoria 0110, South Africa; peter.thompson@up.ac.za (P.N.T.); Abiodun.adesiyun@up.ac.za (A.A.A.); 2Agricultural Research Council–Bacteriology and Zoonotic Diseases Diagnostic Laboratory, Onderstepoort Veterinary Research, Private Bag X 05, Onderstepoort, Pretoria 0110, South Africa; GcebeN@arc.agric.za; 3Agricultural Research Council-Biotechnology Platform, 100 Old Soutpan Road, Onderstepoort, Pretoria 0110, South Africa; PierneefR@arc.agric.za; 4School of Veterinary Medicine, Faculty of Medical Sciences, University of the West Indies, St. Augustine Campus, St. Augustine, Trinidad and Tobago

**Keywords:** LCVs, SCVs, ELISA, tissues, PCR, MLVA

## Abstract

Q fever is a neglected zoonosis in South Africa, causing significant losses in livestock and game animals through reproductive disorders. However, there are limited studies on the extent of *Coxiella burnetii* infections in livestock in South Africa. Further, there is also lack of knowledge about the types of *C. burnetii* strains that are currently circulating in the country. Therefore, a cross-sectional, abattoir-based study was conducted to determine the seroprevalence of *C. burnetii* and associated risk factors, and to characterize *C. burnetii* strains from slaughter livestock at red meat abattoirs in Gauteng, South Africa. Of the 507 animals tested, 6.9% (95% CI: 4.9–9.5%) were positive for antibodies against *C. burnetii*. The seroprevalence was 9.4% (31/331) in cattle, 4.3% (3/69) in sheep, and 0.9% (1/107) in pigs. Out of the 63 tissue samples from 35 seropositive animals including material from two sheep aborted fetuses from Mangaung district (Free State province), 12.7% (8/63) tested positive by IS*1111* PCR. Genotyping of the eight PCR-positive tissues from eight animals by MLVA revealed two novel genotypes, not available in *Coxiella* MLVA databases. It is concluded that slaughter animals pose a risk of exposing abattoir and farm workers to *C. burnetii* in South Africa.

## 1. Introduction

Q fever is caused by the obligate intercellular bacterium, *Coxiella burnetii.* The pathogen belongs to gamma subdivision of Proteobacteria, order Legionellales, family Coxiellaceae and *C. burnetii* is the only species in the genus [1]. The ability to express resistant cell forms enables *C. burnetii* to survive in harsh environments [2]. Once the bacterium enters the phagolysosome of the eukaryotic cell, it undergoes incompletely uncharacterized life cycle forms [3,4]. These are two different morphological forms, namely large (LCVs) and small cell variants (SCVs) [5]. Large cell variants have similar characteristics as typical gram-positive bacteria during exponential growth. These include distinct outer membrane, periplasmic space, cytoplasmic membrane, and diffuse nucleoid, reaching more than 1 µm in length [5]. Compared to LVCs, SCVs are smaller, 0.2 to 0.5 µm in diameter and have electron-dense, condensed chromatin, and condensed cytoplasm. These SCVs are resistant to osmotic shock, oxidative stress, heat shock, sonication, and pressure, one reason that *C. burnetii* is able to withstand harsh environmental conditions [6].

*Coxiella burnetii* infects a wide range of hosts including cattle, sheep, goats, and wild animals such as deer, buffaloes, squirrels, and rabbits [7]. The bacterium is excreted in milk, feces and birth products of infected animals. Animals mainly get infected through inhalation of dried aerosolized particles originating from milk, feces, and birth products such as placenta of infected animals [1].

There is evidence that Q fever infections cause huge economic losses in livestock and wildlife through late abortions, premature births, and low birth weights [8]. For instance, in the Netherlands 50,355, goats and sheep were culled between December 2009 and June 2010 on 89 bulk tank milk positive farms causing huge losses [9]. Furthermore, during the outbreak in the Netherlands, more than 4000 confirmed *Coxiella* human cases were reported with 20% of the infected individuals hospitalized [10]. Despite this evidence, Q fever is not widely known to be endemic in South Africa and thus the disease is not recognized as a controlled and notifiable disease in terms of the Animal Diseases Act 35 of 1984. Thus, there is little known on the prevalence of Q fever in livestock and wildlife in South Africa. The disease goes unnoticed causing huge economic losses through reproductive disorders such as late abortions as it is not continuously monitored.

Despite a study by [11] recently reporting Q fever seroprevalence of 38.4% in cattle in Mnisi community, Bushbuckridge municipality, South Africa, little is known on the prevalence of Q fever in livestock or wildlife in South Africa, albeit in red meat abattoirs. Furthermore, studies by [11,12] were based only on serological diagnosis of Q fever focusing on detection of antibody production. The study by [13] focused on *Coxiella* PCR detection from ticks, and in all the above-mentioned studies, no characterization of the *Coxiella* genotypes was conducted. Thus, there is also no knowledge of genotypes of *C. burnetii* currently circulating in the country. Due to limited data available on Q fever seroprevalence and lack of knowledge on *C. burnetii* strains currently circulating in South Africa, we investigated the seroprevalence and associated risk factors, and further characterized *C. burnetii* strains circulating in slaughter animals at red meat abattoirs in five districts of Gauteng, the most populous province of South Africa, and from a farm in Mangaung district in the Free State province.

## 2. Materials and Methods

### 2.1. Study Area

A serological study was conducted in Gauteng province which is situated in the Highveld region of South Africa with a total area of 18,176 km^2^. The province consisted of six districts, namely City of Tshwane located in the north-eastern part of the province covering an area of 6298 km^2^; City of Johannesburg, 1645 km^2^; Sedibeng district covering most of south-eastern and western Johannesburg with an area of 4173 km^2^; Ekurhuleni district in eastern and north-eastern part of Johannesburg with an area of 1975 km^2^; West Rand, which covers the south-western part of the province with an area of 4087 km^2^; and Metsweding, which was later merged with Tshwane, located in the north-east and western part of the province with total area of 1643 km^2^ (Figure 1). For IS*1111* PCR and MLVA genotyping, we also included pooled organs from two sheep fetuses (liver, spleen, and lungs) which were obtained from a farm in Mangaung district in the Free State province. The Free State province has with an area of 6284 km^2^. A questionnaire was used to obtain demographic information on the animals since they originated from different provinces in South Africa. Other data such as sex, origin of animals (auctions/feed lots), and species were obtained from respective abattoir managers using questionnaires.

### 2.2. Sample Size and Study Population

The sample size for serological tests was estimated using the formula of [15]: n = 1.96^2^ × p × (1 − p)/d^2^ where p = estimated prevalence, d = precision, n = estimated sample size. Considering that there are no current data on the prevalence of Q fever in livestock sampled in red meat abattoirs in South Africa, we used an estimated prevalence of 50% i.e., 0.5 and a precision of 4.5% i.e., 0.045, giving a required minimum sample size of 475. However, for the study, a total of 507 serum samples and 1018 corresponding reproductive tissues were collected at during slaughter at 19 randomly selected red meat abattoirs in Gauteng province (Table 1). These abattoirs consisted of 16 high-throughput (HT) (daily output ≥ 25 carcasses) and three low-throughput (LT) red meat abattoirs in five districts of Gauteng (excluding City of Johannesburg). The 1018 reproductive tissues collected from 507 slaughter animals during slaughter in Gauteng red meat abattoirs comprised 355 penises and 355 testes. A further 80 ovaries, 4 oviducts, 79 mammary glands, and 145 uterus tissue samples were collected from slaughter livestock in Gauteng red meat abattoirs during slaughter (Table 1). In addition, there were also two pooled aborted sheep tissue samples (liver, spleen, and lungs) from diagnostic samples collected from a farm in the Manguang district, Free State province (Table 1). Each abattoir was visited once and based on the study design 30 and 20 serum samples were collected from HT and LT abattoirs respectively, whenever the number of slaughtered animals was available. Blood samples were collected using Bencton, Dickinson and Company (BD)-Vacutainer^®^ SST^TM^ II Advance 10 mL serum collection tubes (Franklin Lakes, NJ, USA) and transported to laboratory at room temperature. Serum was harvested same day by centrifuging the clotted blood in collection tubes at 1000× *g* for 10 min and stored at −20 °C until analyzed. Tissues were collected in zip lock bags and transported to the laboratory the same day at 4 °C to be stored at −20 °C until analysis.

### 2.3. Serological Testing

For detection of IgG antibodies against *C. burnetii*, the IDEXX Q fever 2/strip antibody test kit was used according to manufacturer’s instructions (IDDEX Laboratories, Liebelfld-Bern, Switzerland). All reagents from the IDEXX Q Fever 2/strip antibody test kit together with the frozen serum samples were brought to room temperature and concentrated wash buffer diluted 10 times with distilled water. The diluted buffer was used to dilute the negative, positive controls, as well as the serum samples 400-fold. Exactly 100 µL of the diluted control and collected serum samples were transferred into the *C. burnetii* antigen-coated plate wells and incubated at 37 °C for one hour (h). The plate was then washed three times with 300 µL of the 10-fold diluted wash buffer using a BioTek ELx50 automated microplate washer (BioTEk, Winooski, VT, USA), 100 µL of conjugate was added to each well, and the plates incubated at 37 °C for 1 h. The plate was washed again three times with 300 µL of the 10-fold diluted wash buffer, 100 µL of 3,3′,5,5′-Tetramethylbenzidine (TMB) substrate was then added to each well and the plates incubated at room temperature for 15 min, away from direct light. After 15 min, 100 µL of stop solution was added to each well and the plates immediately read at 450 nm using Thermo Labsystems Multiskan MS Original microplate reader (Thermo Fischer Scientific, Waltham, MA, USA). The assay was validated as follows: For the assay to be valid, the average optical density value of the two negative controls (NCx) at 450 nanometers (A450) should be less than or equals to (≤) 0.500. The average value of the two positive controls (PCx) at 450 nanometers (A450) should be less than or equals to (≤) 2.500. Then PCx-NCx (A450) should be greater than or equals to (≥) 0.300. Sample to positive (S/P) ratio was calculated according to the formula:(1)$$S/P\;\%=100\times \frac{\mathrm{Sample}\left(A450\right)-\mathrm{NCx}\left(A450\right)}{\mathrm{PCx}\left(A450\right)-\mathrm{NCx}\left(A450\right)}$$

For interpretation of results, S/P% < 30% represented a negative result, 30% ≤ S/P% < 40% a suspect, while S/P% ≥ 40% represented a positive result. Seven samples gave suspect results. These samples were repeated once using three different aliquots of the same sample and all of them were negative. The identified 63 reproductive tissues from Gauteng red meat abattoirs corresponding to positive serum samples, including two pooled fetus tissue samples, were subjected to PCR confirmation and MLVA typing.

## 3. Molecular Characterization

### 3.1. DNA Extraction and PCR for Detection of C. burnetii

PCR confirmation was conducted on reproductive tissues from all seropositive animals from this study as well as for diagnostic tissue samples. The tissue samples consisted of 61 tissues from 35 animals (12 mammary glands, 7 uterus, 18 penis, 21 testes, and 3 ovaries) and two pooled aborted sheep fetus tissues from two animals (spleen, liver, and lungs) (Table 1). Tissue samples were cut into small pieces and 10 g from each sample in placed 10 mL ice cold buffered phosphate saline (PBS) pH 7.4 in 50 mL bead ruptor homogenizing tubes containing 2.8 mm ceramic beads. The tissue samples were then homogenized using the automated BEAD RUPTOR ELITE Bead Mill homogenizer (Omni International, Kennesaw, GA, USA). The tissues were homogenized at a speed of three meters per second (3 m/s) for 90 s (s). DNA extraction from the homogenates was conducted using the Qiagen DNeasy^®^ blood and tissue kit as previously described by [16]. The homogenates were centrifuged for 15 min at 4000 rpm and 200 µL of the supernatant transferred to 2 mL centrifuge tubes. To the tubes, 180 µL of tissue lysis (ATL) buffer, 20 µL of proteinase K were added, suspension vortexed, and incubated at 56 °C overnight. After overnight incubation, 200 µL of lysis (AL) buffer was added and the suspension vortexed for 15 sand incubated at 70 °C for 10 min. Absolute ethanol (200 µL) was added to the mixture, vortexed, and transferred to DNeasy^®^ spin columns. The columns were then washed twice with wash buffers; AW1 and AW2, respectively. DNA was eluted from the columns with 200 µL of elution buffer (AE).

PCR for detection of *C. burnetti* in tissues of the seropositive animals was conducted in a 50 µL reaction targeting the multi-copy transposase gene in insertion element; IS*1111* [17] using primers described in Table 2 The *Coxiella* gene fragment (gblock) from Integrated DNA Technologies (Coralville, IA, USA) and water were used as positive and negative controls in the reaction, respectively. The reaction mixture contained 400 nM of each primer as listed in Table 2 (IS*1111*F and IS*1111*R), 25 µL of the Ampliqon 2× Taq DNA polymerase Master Mix Red (Ampliqon A/S, Odense, Denmark) and 10 µL of the extracted DNA. PCR amplification was conducted using BIO RAD T100™ thermal cycler (BIO RAD, Hercules, CA, USA). Cycling conditions consisted of initial denaturation at 95 °C for 15 min, 35 cycles of denaturation 95 °C for 30 s, annealing at 60 °C for 30 s, and extension at 72 °C for 60 s for 35 cycles. Final extension was carried out at 72 °C for 10 min and amplicons visualized on a 1.5% *w/v* ethidium bromide-stained agarose gel with an expected size of 146 bp [17] estimated using Quick-Load^®^ 100 bp DNA Ladder (New England Biolabs, Ipswich, MA, USA). 

### 3.2. Sequence Confirmation of C. burnetii

PCR confirmation of tissues from two animals (n = 2) STCN17 and 40241 from Gauteng province red meat abattoirs and a farm in Mangaung, Free State province, respectively, was conducted using Sanger sequencing. The IS*1111* PCR products of the two tissues were sent to Inqaba Biotechnical industries (Pty) Ltd (Pretoria, South Africa). For sequencing and sequences manually edited using the BioEdit Sequence alignment editor (version 7.2.5). PCR positive DNA from eight seropositive tissue samples from the eight animals (n = 8) was then subjected to MLVA typing. 

### 3.3. MLVA Typing

Eight PCR positive DNA samples from eight animals were genotyped using the Dutch six-locus MLVA panel as previously described by [18,19,20]. This panel utilizes two sets of microsatellite markers. Panel one consists of hexanucleotide repeats (MS27, MS28, and MS34) while panel two is made up of heptanucleotide repeats (MS23, MS24, and MS33) as in Table 2. PCR amplification of the microsatellite markers was carried out in total reaction volume of 20 µL reaction containing 5 µL of the extracted DNA, 10 µL of the Ampliqon 2× Tag DNA polymerase Master Mix Red (Ampliqon A/S, Odense, Denmark. The microsatellite primer pairs are as described in Table 2. The PCR reactions were conducted using BIO RAD T100™ thermal cycler (BIO RAD, Hercules, CA, USA). PCR conditions consisted of initial denaturation at 95 °C for 10 min, 40 cycles of denaturation 95 °C for 30 s, annealing at 60 °C for 30 s, and extension at 72 °C for one min for 40 cycles. Final extension for at 72 °C for 10 min. Microsatellite panel one amplicons visualized on a 2% *w/v* ethidium bromide-stained agarose gel while panel two amplicons were visualized on a 3% *w/v* agarose gel [21,22]. Amplicon sizes for each marker were estimated using O’RangeRuler 20 bp DNA ladder (Thermo Fisher Scientific, Waltham, MA, USA).

**Table 2 pathogens-10-00258-t002:** List of PCR primers used in *Coxiella* IS*11111* PCR and MLVA typing.

Name	Primer Sequence	Reference(s)
**IS*1111F***	5′ CGCAGCACGTCAAACCG3′	[17]
**IS*1111R***	5′TATCTTTAACAGCGCTTGAACGTC3′	[17]
**MS23F**	5′CGCMTAGCGACACAACCAC3′	[18,20]
**MS23R**	5′GACGGGCTAAATTACACCTGCT3′	[18,20]
**MS24F**	5′TGGAGGGACTCCGATTAAAA3′	[18,20]
**MS24R**	5′GCCACACAACTCTGTTTTCAG3′	[18,20]
**MS27F**	5′TCTTTATTTCAGGCCGGAGT3′	[18,20]
**MS27R**	5′GAACGACTCATTGAACACACG3′	[18,20]
**MS28F**	5′AGCAAAGAAATGTGAGGATCG3′	[18,20]
**MS28R**	5′GCCAAAGGGATATTTTTGTCCTTC3′	[18,20]
**MS33F**	5′TCGCGTAGCGACACAACC3′	[18,20]
**MS33R**	5′GTAGCCCGTATGACGCGAAC3′	[18,20]
**MS34F**	5′TTCTTCGGTGAGTTGCTGTG3′	[18,20]
**MS34R**	5′GCAATGACTATCAGCGACTCGAA3′	[18,20]

## 4. Data Analysis

The data were coded using Microsoft Excel and analyzed in Stata 15 (StataCorp, College Station, TX, USA). Exact binomial confidence intervals were calculated for proportions. Univariate associations of sex, species, breed, abattoir throughput, district, and origin of animals with *C. burnetii* seropositivity were assessed using a two-tailed Fisher’s exact test. All variables were then entered into a mixed-effects multiple logistic regression model with abattoir as a random effect; however, due to collinearity with species, breed was not considered in the multivariable model. The model was developed by backward elimination until all remaining variables were significant (Wald *p* < 0.05). Model fit was assessed using the Hosmer–Lemeshow goodness-of-fit test.

*Coxiella burnetii* Sanger sequencing data were analyzed using basic local alignment tool; BLAST; NCBI [23]. MLVA data were analyzed using Bionumerics 7.6 software (Applied Maths, Sint-Martens-Latem, Belgium). Copy numbers were for each marker (MS23, MS24, MS27, MS28, MS33, and MS34) were obtained by extrapolating fragments sizes for each marker with those of the Nine Mile RSA493. From the comparisons of the obtained fragments with the Nine Mile RSA493, a sequence number representing number of repeats for each locus and sample was determined, resulting into the defining of the MLVA profile of the particular sample [24]. The relationship between the genotypes on the study and previously on described genotypes was determined by comparing them with *C. burnetii* genotypes found in the *C. burnetii* 2014 Nijmegen database created by J.J.H.C. Tillburg [24]. Unweighted Pair Group Method with Arithmetic mean (UPGMA) was used to calculate genetic distances between the genotypes and used to generating dendrograms and minimum spanning trees. First, distance matrices were calculated using the “daisy” function with the “gower” parameter specified to determine Gower distances with the R package “cluster” [25]. Thereafter, UPGMA trees were constructed and visualized with ggtree [26]. Minimum spanning trees were calculated using the “ape” package [27] with the “mst” function and visualized using “igraph” [28] and “ggnetwork” [29]. R v.4.0.2 [30] was used for the construction of dendrograms and minimum spanning trees.

## 5. Results

### 5.1. Serology

Of the 507 animals tested, 6.9% (95% CI: 4.9–9.5%) were positive for antibodies against *C. burnetii*. The seroprevalence in livestock by species was 9.4% (31/331) in cattle, 4.3% (3/69) in sheep, and 0.9% (1/107) in pigs. The difference in seroprevalence between sexes was most pronounced in cattle, where it was 17/81 (21%) in females vs. 14/250 (6%) in males (Table 3). Significant univariate associations with *C. burnetii* seropositivity were seen for species, sex, breed, district, and animal origin (Table 3). However, sex and district were no longer significant in the final multivariable logistic regression model (Table 4). The Hosmer–Lemeshow test showed adequate model fit (*p* = 0.670). Animals from auctions were more likely to have been exposed to *C. burnetii* than animals from farms and feedlots (OR = 5.7; 95%CI: 2.6–12.4; *p* < 0.001). Although not significant in the univariate analysis, the multivariable model showed that the odds of Q fever seropositivity in LT abattoirs was significantly higher than in HT abattoirs (OR 4.1; 95%CI: 1.2–14.0; *p* = 0.023) (Table 4). 

### 5.2. Molecular Detection and MLVA Typing of C. burnetii

Molecular detection of *C. burnetii* in 63 tissues from 35 seropositive animals, positive by IS*1111* PCR showed that 12.7% (8/63) were positive with amplicons approximately 146 bp in size (Figure 2). The eight positive tissues comprised 9.5% (6/63) that originated from cattle in red meat abattoirs of Gauteng province while 3.2% (2/63) were pooled organs from aborted sheep fetuses originating from a farm in Mangaung district, Free State province (Table 1).

The edited sequences from two isolates using BLAST revealed that the two amplicons, STFCNC17 and 40241, had a 99.1% similarity with the partial coding sequence of *C. burnetii* strain 54T1 transposase gene (MT268532.1). There was uniform amplification for all the six cattle tissue samples and two sheep samples for markers MS23, MS24, MS27, MS28, and MS33 but only the two sheep samples from Mangaung district amplified for MS34. MLVA profiles of six cattle and two sheep isolates were determined and compared with *C. burnetii* genotypes found in *the C. burnetii* 2014 Nijmegen database created by J.J.H.C. Tillburg found on the database: (http://mlva.u-psud.fr/mlvav4/genotyping/view.php (accessed on 14 January 2021)). The comparisons revealed that the six cattle tissues isolates (STFC17, RTSC8, RNBRC16, KMLDC8, MGC11, and MCM21) from Gauteng red meat abattoirs were novel genotypes, not available on public databases. These genotypes shared a close relationship with a human blood and valve isolate 20090317Frankrijk004 from Marseille, France (Figure 3) with a distance of three map units (m.u). The 2009031Frankrijk004 isolate from Marseille, France, belongs to MLVA genotype 92 and MST 1 and does not originate from an abattoir (Figure 4). The two sheep isolates from Mangaung district (40241 and 40242) were also previously uncharacterized genotypes not available in the public databases. They also shared a close relationship with a human blood and valve isolate 20090317Frankrijk044 from Marseille, France with a distance of three map units (3 m.u; Figure 5). The Frankrijk044 isolate belong to MLVA genotype 92 and MST 1 and does not originate from an abattoir. Minimum spanning trees (Figure 4 and Figure 6) show that by geographic location, all the eight isolates were closely related to human isolates from France, although they were isolated from slaughter livestock.

## 6. Discussion and Conclusions 

This is the first abattoir-based study to show serological evidence of antibodies to *C. burnetii* in livestock using indirect ELISA in South Africa. We observed a seroprevalence of 6.9% with the highest found in cattle and the lowest in pigs. The serology results are consistent with a study conducted by [31] in Northern Ireland. This study reported 64.5% Q fever seroprevalence in dairy cattle as compared to 21% in sheep using indirect immmunofluorescence assay. Similarly, a considerably higher seroprevalence of Q fever (39%) was detected in cattle in Zimbabwe [32] also using the indirect immmunofluorescence assay compared to 9.4% in cattle observed in our study using indirect ELISA. Other studies have found little differences in Q fever seroprevalence between cattle (21.7%) and sheep (28.4%) in the Volta region of Ghana [33], which is consistent with our study as there was no significant difference in seroprevalence between cattle and sheep in the multivariable model (*p* = 0.369).

Another study by [34] reported highest seroprevalence of Q fever in pigs (11.3%) than in both cattle (4.3%) and sheep (0.0) in Trinidad using capillary agglutination test. This is in contrast with findings in the present study where we observed the lowest Q fever seroprevalence in pigs (0.9%) followed by sheep (4.3%) with cattle having the highest prevalence (9.4%). 

In this study, breed differences were noted, with highest seroprevalence (27%) detected in Nguni cattle compared to other breeds. There is therefore a possibility that Nguni breed of cattle may have increased exposure potential for *C. burnetii* in the country and possibly may expose abattoir workers and veterinarians to Q fever. This is because one of the Q fever transmission routes from animals to humans is direct contact with infected carcasses and birth products [1]. 

We observed a higher seroprevalence of *C. burnetii* in females (11.8%) than in males (4.8%), although the differences were no longer significant in the multivariable model (*p* = 0.258). This result is consistent with a study done on Danish cattle by [35], which found a higher true seroprevalence of Q fever in females (9.4%) than in males (2.6%). This the higher Q fever seroprevalence in females than males can be explained in part that an infected male could infect several females during reproductive cycles. *Coxiella burnetii* have been detected in bulls’ semen, indicating sexual transmission of the disease [36].

We observed in the study that livestock originating from auctions were more likely to have been exposed to *C. burnetii* compared to livestock from farms and feedlots. This could be partly explained by the fact that animals purchased from auctions may have originated from a wide range of farms with possible differences in exposure to *C. burnetii*, together with possible close contact of animals in pens at auctions, thus facilitating the spread of the pathogen among animals.

The current data showed statistically significant (*p* = 0.003) differences in prevalence of antibodies against *C. burnetii* between districts where slaughter livestock were sampled. This might not represent the effect of geographical locations of farms and abattoirs in Gauteng province on the prevalence of antibodies to *C. burnetii.* This is because animals slaughtered in abattoirs may originate from other provinces across the country. However, based on the locations of the abattoirs, our findings display differences in seroprevalence of Q fever ranging from 0.0% in the West Rand to 12.2% in Tshwane districts of Gauteng province. The significantly high seroprevalence of Q fever in the then South-Eastern Transvaal (now Mpumalanga) observed by [12] corresponds with a report by [11] who observed a higher Q fever seroprevalence in the Bushbuckridge area of Mpumalanga province as compared to the findings in the study. Similarly, Ref. [12] reported a significantly higher seroprevalence of Q fever in the then South-Eastern Transvaal (now Mpumalanga) than in the then Western Transvaal (now North West Province), which was attributed to the differences in the distribution of the blue tick, *Boophilus decoloratus*. This tick was a known important vector of the pathogen in 1985, which was a dry year [12]. During that time in 1985, the abundance of this tick was low in the Western Transvaal corresponding with low Q fever seroprevalence in that area. This may suggest that *B. decoloratus* is involved in the transmission of Q fever. Another tick species, *Haemaphysalis leachi*, a known Q fever carrier in South Africa had a distribution closely like that of Q fever in the same year, 1985 [12]. This suggests that this tick species might also be involved in Q fever transmission among cattle in Gauteng province. 

We observed Q fever prevalence of 12.7% by PCR in the study. This is relatively low compared to other studies such as [13] who reported 41% prevalence. Other similar studies have also reported higher Q fever prevalence elsewhere using PCR. For instance, Ref. [22] reported *C. burnetii* prevalence of 16.6% from ruminants and wildlife in Portugal using IS*1111* PCR. In the study by [37], goats had the highest prevalence with 23.5%, cattle with 20.8%, and sheep with 10%. These observations by [37] differ with the findings in our study where cattle reported highest Q fever prevalence with 9.5% (6/63) while sheep reported 3.2% (2/63) using IS*1111* PCR.

Molecular characterization of the *C. burnetii* isolates using MLVA showed that the eight isolates are novel genotypes. However, they are closely related to genotypes from France, Europe, albeit human isolates, with genetic distance of 3 m.u. This could suggest that two genotypes were discovered, one that could be specific to cattle and the other to sheep. This observation also suggests that the isolates from this study in South Africa and the ones from France might have originated from different locations but shared a common ancestor. 

Based on the seroprevalence and PCR detection of Q fever, cattle had the highest level of exposure to the pathogen compared to sheep and pigs. There is also the possibility that cattle pose the highest risk of abattoir workers being exposed to the pathogen at slaughter and processing. Although the prevalence was not high, the detection of widespread exposure to *C. burnetii* in slaughter livestock cannot be ignored. Hence, the pathogen may be of economic importance to the livestock industry and of zoonotic significance to personnel in the farming sector, including livestock farmers, animal attendants, and veterinarians, and to consumers of under-cooked products such as milk. It is therefore imperative to conduct more studies on Q fever in livestock in other provinces of South Africa. These studies should specifically focus on the isolation and molecular, genomic, and proteomic analyses of circulating strains of *C. burnetii* in the country. 

In conclusion, we have documented the seroprevalence and risk factors associated with *C. burnetii* in red meat abattoirs of Gauteng province, South Africa. In addition, we have detected *C. burnetii* by PCR in slaughter animals as well as from aborted fetuses. Using MLVA we detected two novel genotypes in sheep and cattle, respectively. 

## Figures and Tables

**Figure 1 pathogens-10-00258-f001:**
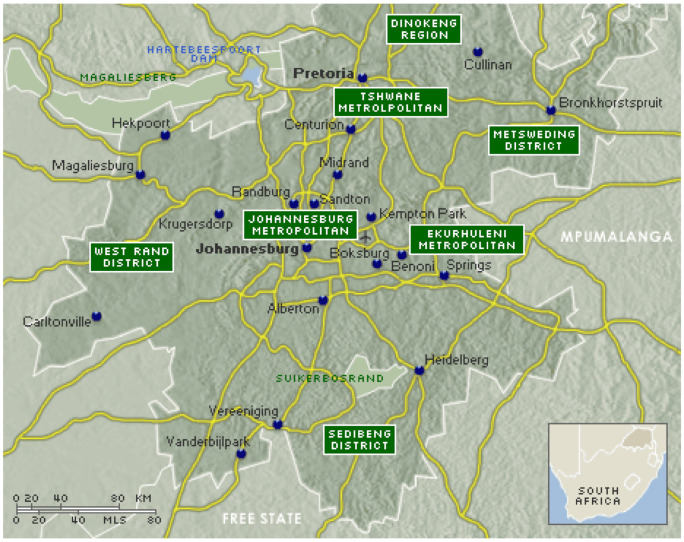
Map of Gauteng province districts where blood and reproductive tissues were collected from red meat abattoirs [14].

**Figure 2 pathogens-10-00258-f002:**
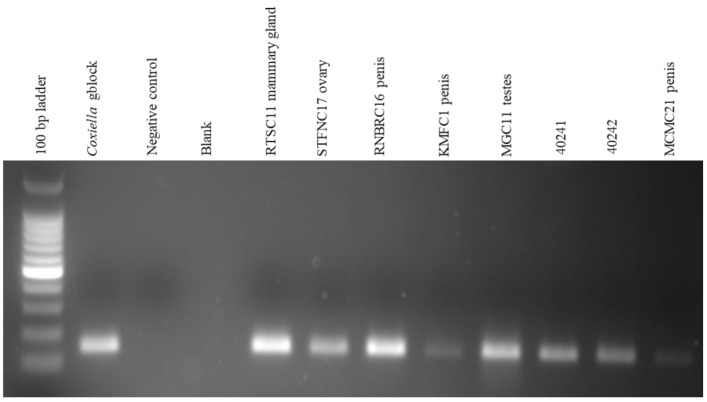
Gel electrophoresis of seropositive tissue DNA Using IS*1111* PCR. The first lane is Quick-Load^®^ 100 bp DNA Ladder (New England Biolabs, Ipswich, MA, USA). The positive control is *Coxiella* gene fragment (gblock) from IDT (Coralville, IA, USA), negative control is distilled water and blank is an empty lane. RTSC11 Mammary gland, STFNC17 ovary, RNBRC16 penis, KMFC1 penis, MGC11 testes, and MCMC21 penis are cattle samples from Gauteng red meat abattoirs while F40241 and 40242 are pooled sheep fetuses samples from a farm in Mangaung district. The expected PCR product is approximately 146 bp as shown in the figure.

**Figure 3 pathogens-10-00258-f003:**
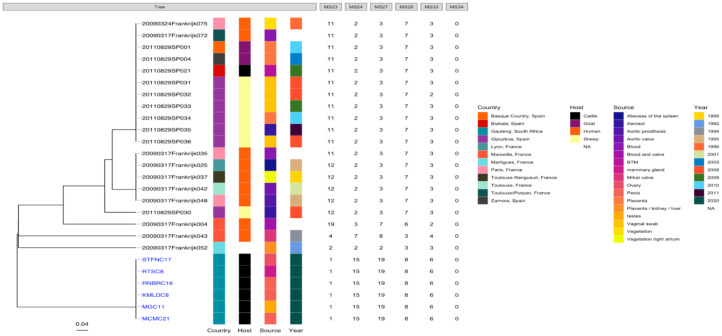
Phylogenic tree of *C. burnetii* genotypes of six tissues from six animals (STFC17, RTSC8, RNBRC16, KMLDC8, MGC11, and MCM21) originating from Gauteng red meat abattoirs based on MLVA-6 database (http://mlva.u-psud.fr/mlvav4/genotyping/view.php (accessed on 14 January 2021)) created by Dr J.J.H.C. Tillburg. The genotype did not amplify for all tissues for MS34 (data not shown) and Nine Mile RSA493 with copy numbers 9-27-4-6-9-5 was used reference (data not shown).

**Figure 4 pathogens-10-00258-f004:**
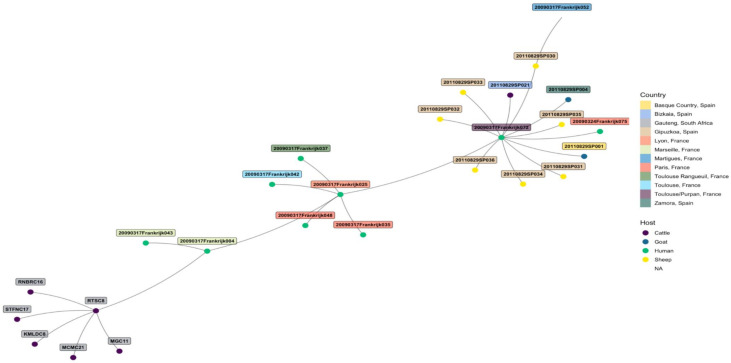
Minimum spanning tree showing the relationship between the eight tissue samples from Gauteng tissue samples from Gauteng province red meat abattoirs and genotypes from the *C. burnetii* 2014 Nijmegen database by geographical location show that the samples are closely related with human isolate 20090317Frankrijk004 from Marseille, France. The eight tissues from Gauteng province red meat abattoirs share the same MLVA-6 copy numbers as the isolate 20090317Frankrijk004.

**Figure 5 pathogens-10-00258-f005:**
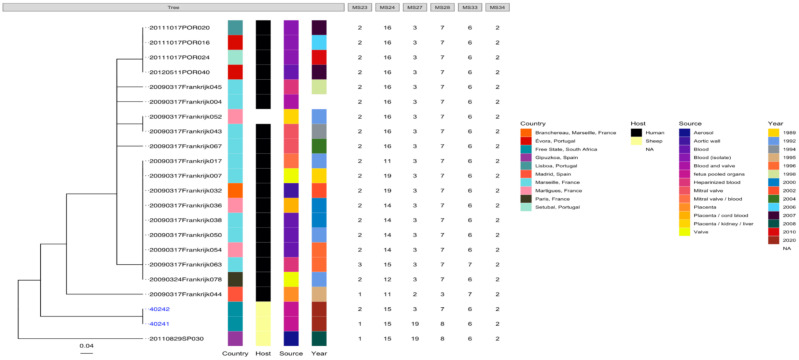
Phylogenic tree of *C. burnetii* genotypes of two sheep tissues (pooled organs) from two animals (40241 and 40242) originating a farm in the Mangaung district based on MLVA-6 database (http://mlva.u-psud.fr/mlvav4/genotyping/view.php (accessed on 14 January 2021)) created by Dr J.J.H.C. Tillburg. This genotype amplified for all MLVA-6 markers Nine Mile RSA493 with copy numbers 9-27-4-6-9-5 was used reference (data not shown).

**Figure 6 pathogens-10-00258-f006:**
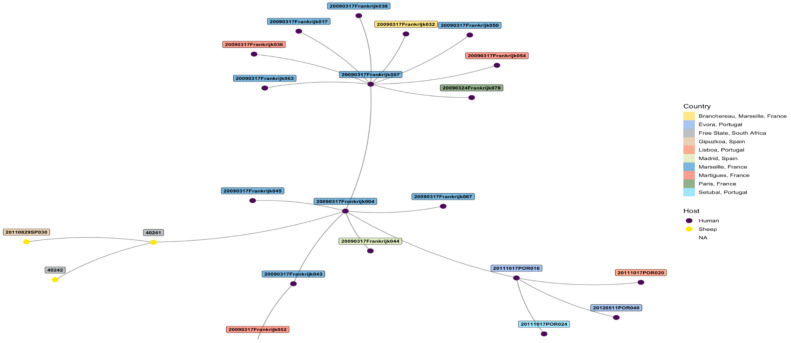
Minimum spanning tree showing the relationship between the two sheep tissue samples (40241 and 40242) from a farm in Mangaung district and genotypes from the *C. burnetii* 2014 Nijmegen database by geographical location show the isolates samples are closely related with human isolate 20090317Frankrijk044 from Marseille, France.

**Table 1 pathogens-10-00258-t001:** Types of samples collected at red meat abattoirs in Gauteng province and tissue samples from a farm in Free State province used in the study.

Province	Sample Type	Sample Name	Number (n)
Gauteng	Serum	Serum	507
Gauteng	Tissue	Penis	355
Gauteng	Tissue	Testes	355
Gauteng	Tissue	Uterus	145
Gauteng	Tissue	Mammary gland	79
Gauteng	Tissue	Ovary	80
Gauteng	Tissue	Oviduct	4
Free state	Pooled tissues	Spleen, liver, lung	2

**Table 3 pathogens-10-00258-t003:** Seroprevalence of Q fever in livestock slaughtered at red meat abattoirs in Gauteng, South Africa, and univariate analysis of associated factors.

Variable	Level	Prevalence (%)	95%CI*	*p*-Value
**Species**	Bovine	9.4	6.5–13.0	0.003
	Ovine	4.3	0.9–12.0	
	Porcine	0.9	0.002–5.0	
**Sex**	Male	4.8	2.8–7.6	0.007
	Female	11.8	7.2–18.1	
**Breed**	Bonsmara	6.6	3.8–10.5	<0.001
	Jersey	6.9	1.4–19.0	
	Nguni	26.7	14.7–42.0	
	Dorper	4.3	0.9–12.0	
	Large white	0.9	0.2–5.0	
**District**	Tshwane	12.2	7.9–17.8	0.003
	Ekurhuleni	3.4	0.7–9.7	
	Metsweding	10	1.2–32.0	
	Sedibeng	4.3	1.7–8.7	
	West Rand	0	0.0–7.1	
**Abattoir throughput**	High	7.1	4.8–9.9	1
	Low	5.8	1.6–143.9	
**Origin of animals**	Farm/feedlot	4.8	2.9–7.3	<0.001
	Auction	16.7	9.6–26.0	
**Total**		6.9	4.9–9.5	

CI* Confidence interval.

**Table 4 pathogens-10-00258-t004:** Multivariable analysis of risk factors associated with seropositivity to *C. burnetii* in livestock at red meat abattoirs in Gauteng, South Africa.

Variable	Level	Odds Ratio	95% CI*	*p*-Value
**Species**	Bovine	1 *	0.2–2.0	0.369
	Ovine	0.6	0.006–0.4	0.003
	Porcine	0.04		
**Abattoir throughput**	High	1 *	1.2–14.0	0.023
	Low	4.1		
**Origin of animals**	Farm/feedlot	1 *	2.6–12.4	<0.001
	Auction	5.7		

CI* Confidence interval. * Reference level.

## Data Availability

Sequence confirmation of *Coxiella* amplicon sequences in the study was conducted by comparing the sequences with those available on BLAST. This information can be found on https://blast.ncbi.nlm.nih.gov/Blast.cgi? (accessed on 14 January 2021) The *Coxiella* MLVA Publicly available datasets were used for comparisons with genotypes discovered in the study. This data can be found on http://mlva.u-psud.fr/mlvav4/genotyping/view.php (accessed on 14 January 2021).

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
