# Peer review of "Q Fever: Seroprevalence, Risk Factors in Slaughter Livestock and Genotypes of Coxiella burnetii in South Africa"

_pathogens, 2021, doi:10.3390/pathogens10030258_

Round 1

Reviewer 1 Report

I read the manuscript with great pleasure, the work was thoroughly and meticulously prepared, sometimes the methods were even described in too detail, but in principle it does not bother me, and this can only facilitate the reproduction of the research. The research uses adequate techniques and advanced statistical methods, giving interesting and, in my opinion, scientifically valuable results. The aim of the work, in my opinion, touches on a very important problem of epidemic threats from poor countries in Africa or Asia. Research results indicate the emergence of new C. burnetii variants and evidence of their phylogenetic relatedness and epidemic pathways. I only have a few minor comments. Authors should:

1.explain the abbreviations used in the formula (line 120-121)

2.explain how many times the repetition was performed (line 127)

3.check if the reference to other sources in the text is correct, it reads a bit strange that, for example, "described by [12]" without mentioning at least the author's name

The discussion is the weakest part of the work, it gives the impression of being written by a different author than the rest of the manuscript, it is hard to concentrate while reading, some parts are  written quite chaotic or lengthily, the factual and grammatical errors are present e.g. line 42-50, line 62-63, line 65-66 etc. I recommend redrafting this part of the work.

Also:

Line 9-12: this sentence is unclear, the numbers do not indicate "the higher prevalence " or even "the higher differences" in the authors' study in comparison to other publications

Line 18: Do you have any hypothesis regarding the greater prevalence of seropositive individuals in the Nguni cattle compared to others? Can they be individual features or maybe the influence of environmental factors? Maybe it is worth to consider to genotyping this group of animals, maybe there is a specific polymorphism predisposing to the infections?

Sincerely,

Reviewer

Author Response

Dear Reviewer

Good afternoon, please see attachment.

Reviewer 2 Report

I think your paper is worth publishing, still, I have a problem with a little sloppy style of the article. I marked the most blatant errors and offered a corrected form in the comments I made in the enclosed pdf-file (made via Adobe Reader).

Keywords section must be totally altered.

The typescript should be thoroughly analyzed for, especially, double (or more) spaces, commas (or lack of them), units unification (e.g. second i 's', not 'sec') and dividing too long sentences.

Number from 0 to at least 9 should be written zero, one, two...

You mention samples origin too many times.

Please unify - Q fever or Q-fever?

I would also add at least authors' names before numbers indicating references (or the current form is required by the Journal?)

Reviewer 3 Report

The manuscript presents observational data on the incidence of Coxiella burnetti circulating in livestock in South Africa. The study includes serological, PCR and bioinformatic analysis of recovered sera, tissues and Coxiella genetic material. Overall the study has merit but can be improved upon with editing to enhance clarity to the reader. 

See below for overall and then specific comments.

Overall comments:

Authors have missed including a brief description of the coxiella lifecycle in the introduction. This would greatly enhance background information to the reader. Please consider adding.

The authors start with Table 1 and then move onto Table 4? And then the authors move onto Tables 2,3.  Tables should be presented and discussed in the order (Tables 1, 2, 3, 4).

Figures are also discussed out of order. Authors refer to figure 1, 2 and then 4.Then there is a brief mention of figures 3 and 5 together in one sentence. The results shown in Figures 3 and 5 are barely even discussed in the results?  The authors need to reorder their figures to be presented sequentially and should expand description of figures 3 and 5.

Lines 63-77: Authors should consider adding a geographical map with locations of sample collection noted as many readers are not familiar with the geographical area of South Africa. This would be more helpful than just the description of the area in Km^2.

Specific comments

Line 37: loses should be "losses"

Lines 98-100: Can the authors clarify how the samples were stored after collection prior to centrifugation and freezing?  Where they transported and if so, how?

Lines 120-121: Define NCx and PCx

Lines 127-130: This paragraph is confusing. Authors start discussing suspect samples and then jump to positives in the same sentence.  Separate these into 2 sentences. Also, how many of the suspect cases were recategorized into negative or positive after being repeated?

Line 151: grammar regarding "buffers wash buffers"

Lines 153-164: Positive and negative PCR controls are not discussed but do show up in figure 1. Please add to methods.

Line 177: grammar regarding "Panel 1 one consists..."

Line 239: Do the authors mean 12.7 %? (the percent sign is missing)

Figure 1: Orient the figure by rotating 90 degrees so that the gel is seen with bands being horizontal not vertical. 

Figure 1 legend. Define gblock. This reviewer is assuming you mean genomic DNA?  What strain is the positive control from? This was also missing in the methods.  Also, what company was the ladder from?  Please update information to address this in the figure 1 legend (Lines 266-271)

Author Response

Please see attachmnet

Round 2

Reviewer 3 Report

My concerns have been addressed and I am satisfied with the changes. I am endorsing publication.